# A Three-Phase Model to Evaluate Effects of Phase Diffusivity and Volume Fraction upon the Crack Propagation in Concrete Subjected to External Sulphate Attack

Chaofan Yi [1,2] , Zheng Chen [1], Jiamin Yu [1] and Vivek Bindiganavile [2,*]

1   Key Laboratory of Disaster Prevention and Structural Safety of China Ministry of Education,
    School of Civil Engineering and Architecture, Guangxi University, Nanning 530004, China
2   Department of Civil and Environmental Engineering, University of Alberta, Edmonton, AB T6G 1H9, Canada
*   Correspondence: vivek@ualberta.ca

**Abstract:** This study models concrete as a multi-phase system that comprises the mortar, coarse aggregates, and interfacial transition zones (ITZs). The diffusivity and the volumetric fraction of these phases are considered to propose a three-phase diffusion–reaction model to simulate the external sulphate attack. Furthermore, the parametric analysis alongside the sensitivity analysis is carried out to quantify the effect of these phases on the expansive cracking in concrete when exposed to a sulphate-rich environment. The results show that mortar dominates the sulphate ingress and the ensuing distress, while the ITZ is found to be least significant. Due to its significantly low permeability, the coarse aggregate may act as a "deceleration strip" or a "dam", which in turn obstructs the sulphate penetration. More importantly, this effect is further noted to evolve with a decrease in the diffusivity and a rise in the volumetric fraction of coarse aggregates. As for ITZ, its volume fraction may play a more dominant role than its diffusivity on sulphate attack in concrete.

**Keywords:** three-phase model; diffusion–reaction; coarse aggregate; interfacial transition zone; external sulphate attack

## 1. Introduction

In part due to the easy availability of its raw materials and largely due to its numerous thermo-mechanical attributes, Type GU Portland cement (CSA), also called ordinary Portland cement (OPC), and its composites have become the most popular building material, with extensive applications in construction. However, the active nature of hydration products implies that the associated cementitious system is susceptible to various chemical environments [1]. In particular, external sulphate attack remains one of the most complex and potentially dangerous concerns. Accordingly, extensive interest has been directed toward the mechanisms underlying the external sulphate attack and the ensuing distress in concrete [2–5]. Prior studies revealed that such a chemical attack starts with the physical diffusion of sulphate ions, accompanied by a series of chemical reactions between the penetrating sulphate ions and the inherent calcium aluminate phases. This subsequently leads to formations of gypsum and/or ettringite [6–10], both of which expand in volume to densify the concrete microstructure temporarily [11]. This phenomenon may sometimes be useful for "self-healing" [12]. However, further expansions beyond the tensile capacity of cementitious materials will cause the onset and propagation of cracks within concrete [11]. The phenomenon eventually leads to the noticeable reduction in strength and greater permeability that aids in the transport of deleterious substances.

Considering the chemical activity of OPC, its chemical composition has been reported to dominate the sulphate attack in concrete systems. Specifically, the lower the calcium aluminate content, the better the sulphate resistance [12]. Furthermore, a decrease in the water-to-binder ratio and an extension in the curing duration were both recognized to

refine the pore structure, which in turn improved the resistance of concrete to sulphate ingress [13–15]. The ionic concentration was also noted to play a part. According to diffusion kinetics, the higher environmental concentration usually translates into a stronger diffusion gradient, which drives the sulphate ions to a deeper location within an equal exposure duration. In addition, recent studies documented that the sulphate-induced deterioration would be affected by other factors. For example, the coarse aggregates embedded in concrete could obstruct the sulphate diffusion and reaction effectively [12,16]. More importantly, the larger size and the greater dosage of coarse aggregates together led to the stronger hindrance in this regard [16]. Furthermore, the presence of other ions, such as chlorides, may trigger a competitive antagonism with sulphate ions, which in turn deters the penetration of sulphate ions [17,18].

Alongside experimental investigation, numerical work helps to forecast the external sulphate attack inside cement and concrete composites. As mentioned above, the external sulphate attack involves both physical diffusion and chemical reactions. Hence, the establishment of a diffusion–reaction model for sulphate attack must take these two behaviors into account. Based on Fick's second law and reaction kinetics, Tixier and Mobasher [19,20] were first in proposing a static diffusion–reaction model. Most subsequent advancements are proposed accordingly [21–27]. In this regard, the effect of chemical activity was considered in Marchand's model [21–23], while Ikumi et al. took the evolving microstructure into account when modelling this durability issue [24]. Zuo et al. took note of the ionic concentration and the electrolyte friction as impacting the time-variant diffusion coefficient of sulphate ions [25]. Sun et al. modelled concrete as an imperfect system and considered the influence of inherent damage on the diffusion behavior of sulphate ions [26]. Similarly, Idiart et al. proposed a damage-based model to account for the role of micro-cracks in the sulphate diffusion–reaction behavior, as per the theory of chemo-mechanics [27]. However, most prior numerical studies do not take the diffusivity and volume fraction of all three phases, i.e., the mortar, coarse aggregate, and interfacial transition zone (ITZ), into account. It is acknowledged that the coarse aggregate deters the sulphate penetration inside cementitious materials [12,16]. Furthermore, a related investigation reported that, compared to the mortar, the coarse aggregate registered a significantly smaller diffusion coefficient of chemical ions. On the other hand, the corresponding ITZ phase had a greater value [28,29]. Furthermore, the aggregate embedded in concrete predominantly comprises crystalline silica, with very few reactive calcium aluminates (CA). As a result, the sulphate attack may only exhibit its diffusion behavior therein. Furthermore, the ITZ as the third phase has of late been taken into account in some recent studies [30,31].

The present study constructs concrete as a three-phase system constituting the mortar, coarse aggregates, and ITZs. The diffusivity and the volumetric fraction of each phase are considered and then lumped as per the Voigt and Reuss theory, when establishing the three-phase diffusion–reaction model. In this manner, the proposed model is able to capture the influence of each individual phase on the diffusion–reaction progress of sulphate ions in concrete. The numerical results are thereafter compared with the experimental data to validate the feasibility of the proposed model. In what follows, the numerical outputs in terms of sulphate concentration and ettringite content are fed to the volumetric expansion models and a durability-based limit state function, in order to forecast the sulphate-induced crack propagation in concrete. Furthermore, a sensitivity analysis is carried out to assess the significance of various phases on crack propagation in concrete structures when exposed to a sulphate-rich environment.

## 2. Model Description

### 2.1. Underlying Chemical Mechanisms

Their active nature and porous character leads to the susceptibility of cementitious systems when employed in a sulphate-rich environment. The penetrating sulphate ions react with the inherent Portlandite (CH) in the presence of water (H) to form an intermediate product, i.e., gypsum. This expansive substance will undergo a series of ensuing reactions

with inherent calcium aluminate phases, including the unhydrated cement ($C_3A$), the tetracalcium aluminate ($C_4AH_{13}$), and the monosulfate ($C_4A\bar{S}H_{12}$). Eventually, they are transformed into ettringite, which is a stable and highly expansive compound. The above chemical process could be variously described by Equations (1)–(3), when the associated aluminate phase is $C_3A$, $C_4AH_{13}$ and $C_4A\bar{S}H_{12}$, respectively. Equations (1)–(3) are as follows:

$$C_3A + 3C\bar{S}H_2 + 26H \rightarrow C_6A\bar{S}_3H_{32} \tag{1}$$

$$C_4AH_{13} + 3C\bar{S}H_2 + 14H \rightarrow C_6A\bar{S}_3H_{32} + CH \tag{2}$$

$$C_4A\bar{S}H_{12} + 2C\bar{S}H_2 + 16H \rightarrow C_6A\bar{S}_3H_{32} \tag{3}$$

These three separate reactions could be combined further into a global form in Equation (4) [19]. To this end, two weighted average stoichiometric coefficients, $\lambda$ and $n$, are introduced, which represent the required gypsum and water, respectively, in proportion to the equivalent calcium aluminate phase (CA). According to the associated Equations (1)–(3), the upper and lower bounds for $\lambda$ and $n$ may be computed as 2~3 and 14~26, respectively, using the weighted average algorithm. Note that this lumped expression has been widely referred to by later studies [25,27] due to its convenience in numerical simulation. Equation (4) is as follows:

$$CA + \lambda C\bar{S}H_2 + nH \rightarrow C_6A\bar{S}_3H_{32} \tag{4}$$

### 2.2. Diffusion–Reaction Model

The preceding studies have broadly confirmed the applicability of Fick's second law when simulating the physical diffusion of chemicals in cementitious composites [32,33]. In sulphate attack, the sulphate diffusion is also accompanied by a series of synchronous chemical reactions. This leads to the consumption of penetrating sulphate ions and the inherent calcium aluminates. While this chemical behavior may reduce the concentration of free sulphate ions instantly, the associated concentration gradient is magnified, which in turn enhances the subsequent diffusion impetus. Therefore, the reaction kinetics must be coupled with Fick's second law. This eventually proposes the combined diffusion–reaction model for sulphate attack in concrete, as is presented in the following Equation (5):

$$\begin{cases} \frac{\partial U}{\partial t} = \frac{\partial}{\partial x}\left(D_u(x,y,t)\frac{\partial U}{\partial x}\right) + \frac{\partial}{\partial y}\left(D_u(x,y,t)\frac{\partial U}{\partial y}\right) - kUC \\ \frac{\partial C}{\partial t} = -\frac{kUC}{\lambda} \\ U(\Omega,0) = 0, C(\Omega,0) = C_0, U(\Gamma,t) = U_0, C(\Gamma,t) = 0, \end{cases} \tag{5}$$

where $U$ and $C$ denote, respectively, the concentration of free sulphate ions and the content of the remaining calcium aluminate phase in the cement composite, mol/m$^3$; $D_u$ represents the sulphate diffusion coefficient, m$^2$/day; $k$ indicates the chemical reaction coefficient between sulphate ions and calcium aluminate phases, equal to $1.0 \times 10^{-7}$ mol/m$^3$·s [16]. The variables '$x$' and '$y$' are the spatial coordinates in a two-dimensional domain, mm, and $t$ is the exposure duration, days. Furthermore, $\Omega$ refers to the entire diffusion–reaction domain in concrete, and $\Gamma$ defines the two-dimensional boundary exposed to the external sulphate source. In addition, the environmental sulphate concentration and the initial calcium aluminate content in the binder are defined as $U_0$ and $C_0$, mol/m$^3$, respectively. Equation (5) was pioneered by Tixier and Mobasher [19,20], and afterwards used in related numerical studies [24–27].

### 2.3. Three-Phase Model

Hanshin and Shtrikman pioneered a two-phase model to compute the elastic modulus of concrete by using the variational theory [34]. Additionally, the Voigt and Reuss models were proposed to find the theoretical upper and lower bounds based on the following two assumptions, respectively: (i) that the axial deformation or strain in the unit interval is

constant, and (ii) that the force across a unit region is the same in two separate phases. The most important backbone for the above two assumptions is the linear correlation between stress and strain ($\sigma = E\varepsilon$). Note here that Fick's first law also reveals a similar linear correlation between the ionic flux and the concentration gradient, as seen in Equation (6). Given this, Hobbs was the first to introduce the above two-phase theory into durability research, with the analogy between the constitutive law and Fick's first law [35]. In this manner, the ionic diffusion coefficient ($D$), the ionic flux ($Q$), and the concentration gradient ($dc/dx$) in Equation (6) are, respectively, equivalent to the elastic modulus ($E$), stress ($\sigma$), and strain ($\varepsilon$) in the constitutive law. The two assumptions adopted in the Voigt and Reuss models bound the theoretical elastic modulus and also the hereto diffusion coefficient. The first assumption manifests in a series model while the second assumption reflects a parallel model. The upper and lower bounds are captured per Equations (7) and (8), respectively. However, the above two equations could only simulate the ideal situation. Hence, Equation (9) is presented to compute the actual diffusion coefficient for concrete, comprising the mortar and coarse aggregates [35]. Thus, the output from Equation (9) varies between the corresponding upper and lower bounds generated by Equations (7) and (8), and also has been confirmed to fit the actual value. Equations (6)–(9) are as follows:

$$Q = -D\frac{dc}{dx} \tag{6}$$

$$D_{upper} = \sum_{i=1}^{2} f_i D_i = f_p D_p + f_a D_a \tag{7}$$

$$D_{lower} = \sum_{i=1}^{2} f_i \frac{1}{D_i} = \frac{D_p D_a}{f_a D_p + f_p D_a} \tag{8}$$

$$D_{p-a} = \frac{(D_a - D_p)f_a + (D_p + D_a)}{(D_p + D_a) + (D_p - D_a)f_a} D_p \tag{9}$$

where $D_c$, $D_a$, and $D_p$ denote the diffusivity of concrete, the coarse aggregate, and the mortar, respectively. Note here that the value of $D_a$ may variously range from 0.1 to 0.001 times that of $D_p$ [29]. Here, $D_{upper}$ and $D_{lower}$ indicate the corresponding upper and lower bounds, respectively, while $f_a$ and $f_p$ represent the volumetric fraction of coarse aggregate and mortar in concrete, respectively, which could be easily obtained from the mix design.

However, it should be noted that the above two-phase model shown in Equation (9) ignores the diffusivity of ITZ. In order to solve this issue, concrete is preliminarily considered as a two-phase composite that only comprises the mortar and coarse aggregates, and the computed diffusion coefficient, $D_{p-a}$, is treated as an intermediate parameter. Subsequently, the mortar–coarse aggregate phase is treated as a combined phase while the ITZ is introduced as the new phase. Once again, the two-phase theory alongside the Voigt and Reuss models is applied to establish the three-phase model for sulphate diffusion, as shown in Equations (10) and (11). Furthermore, a hyperbolic model is introduced here to account for the influence of the evolving micro-structure on the sulphate attack in concrete, as in Equation (12). Equations (10)–(12) are as follows:

$$D_u = \frac{(D_{ITZ} + D_{p-a}) + (D_{ITZ} - D_{p-a})f_{ITZ}}{(D_{ITZ} + D_{p-a}) + (D_{p-a} - D_{ITZ})f_{ITZ}} D_{p-a} \tag{10}$$

$$D_u = \frac{[D_{ITZ} + \frac{(D_a+D_p)+(D_a-D_p)f_a}{(D_a+D_p)+(D_p-D_a)f_a}D_p] + [D_{ITZ} - \frac{(D_a+D_p)+(D_a-D_p)f_a}{(D_a+D_p)+(D_p-D_a)f_a}D_p]f_{ITZ}}{[D_{ITZ} + \frac{(D_a+D_p)+(D_a-D_p)f_a}{(D_a+D_p)+(D_p-D_a)f_a}D_p] + [\frac{(D_a+D_p)+(D_a-D_p)f_a}{(D_a+D_p)+(D_p-D_a)f_a}D_p - D_{ITZ}]f_{ITZ}}[\frac{(D_a+D_p)+(D_a-D_p)f_a}{(D_a+D_p)+(D_p-D_a)f_a}]D_p \tag{11}$$

$$D_p = D_{\min} + (D_{p0} - D_{\min}) \frac{e^{-\beta_D} \cdot \varphi(x,y,t)}{\varphi_0 + (e^{-\beta_D} - 1)\varphi(x,y,t)} \tag{12}$$

where $D_u$ is the three-phase diffusion coefficient for sulphate ions; $D_{ITZ}$ denotes the diffusivity of ITZ, typically ranging from 1.3 to 16.2 times $D_p$ [29], while $f_{ITZ}$ is the volumetric fraction of ITZ. Furthermore, $D_{p0}$ represents the initial diffusion coefficient of the mortar, which has been recognized as $1.0 \times 10^{-12}$ m²/s [24], while $D_{\min}$ denotes the diffusion coefficient of the mortar when the internal capillary pores are completely filled by the resulting ettringite, equal to $D_0/10$ [24,34]. Here, $\beta_D$ is the shape factor, recommended as 1.5 by Idiart [27]. The porosity that evolves with the progress of sulphate attack is given by $\varphi(x,y,t)$, and the initial porosity is defined as $\varphi_0$.

*2.4. Sulphate-Induced Tensile Strain*

Recall from Equations (1)–(3) that the reactions between calcium aluminate phases and the sulphate ions yield a common product, i.e., ettringite, which is known to expand in volume. This expansion leads to a pore-filling effect that densifies the microstructure and, thereafter, causes inevitable cracks once the expansion exceeds the tensile capacity of concrete. The magnitude of volumetric change led by ettringite may be calculated per the difference in specific gravity, as in Equation (13). Then, the volumetric change of the total mixture is determined, thusly, by Equation (14). Equations (13) and (14) are as follows:

$$\begin{cases} \dfrac{\Delta V_{CA_i}}{V_{CA_i}} = \dfrac{(m_V^{C_6A\bar{S}_3H_{32}})^{-1}}{(m_V^{CA_i})^{-1} + \lambda_i (m_V^{C\bar{S}H_2})^{-1}} - 1 \\[2mm] m_V^k = \dfrac{\rho^k}{M^k} \end{cases} \tag{13}$$

$$\left(\frac{\Delta V}{V}\right)_{CA} = \sum_{i=1}^{3} \frac{\Delta V_{CA_i}}{V} = \sum_{i=1}^{3} \frac{\Delta V_{CA_i}}{V_{CA_i}} \frac{V_{CA_i}}{V} = 0.44 \frac{V_{CA_1}}{V} + 0.54 \frac{V_{CA_2}}{V} + 1.32 \frac{V_{CA_3}}{V} \tag{14}$$

where the density, molar mass, and molarity of a certain compound, $k$, are defined as $\rho^k$, $M^k$, and $m_V^k$, respectively. Here, $\Delta VCA_i/VCA_i$ indicates the volumetric variation of ettringite sourced from $CA_i$ (here, $CA_1 = C_4AH_{13}$; $CA_2 = C_4A\bar{S}H_{12}$; $CA_3 = $ anhydrous $C_3A$). Furthermore, $V_{CAi}/V$ represents the fraction of calcium aluminate phase in binder, which could be found as per the following Equation (15) [36]:

$$\frac{V_{CA_i}}{V} = \frac{C_{CA_i} M_{CA_i}}{\rho_{CA_i}} \tag{15}$$

where $C_{CA_i}$, $M_{CA_i}$, and $\rho_{CA_i}$ signify the molar concentration (mol/m³), the molar mass, and the density of each calcium aluminate phase ($CA_i$) [36].

Besides the volumetric variation of ettringite, the expansive strain, $\varepsilon_V^0$, is also dependent on the reacted calcium aluminates, $C_r$, in proportion to the initial content, $C_0$. Accordingly, the average $\varepsilon_V^0$ is expressed as follows in Equation (16) [16]:

$$\varepsilon_V^0(x,y,t) = \frac{C_r(x,y,t)}{C_0} \left(\frac{\Delta V}{V}\right)_{CA} \tag{16}$$

In Equation (16), the reacted calcium aluminate at a given depth, $C_r(x, y, t)$, could be computed as the difference between the initial calcium aluminate content, $C_0$, and the remaining content, as in the following Equation (17):

$$C_r(x,y,t) = C_0 - C(x,y,t) \tag{17}$$

As is well known, cement and concrete composites are porous materials. This means that the intrinsic micro-pores will accommodate a certain amount of expansion led by ettringite, without causing any expansive strain. In addition, ettringite is a needle-like product in shape, which means the corresponding expansion may predominantly proceed

along a "principal" direction. As a result, those inherent pores may not be fully filled when the sulphate-induced strain is noted. Hence, a discriminant function, expressed as Equation (18), is adopted to explain the above latent period and to adjust the actual volumetric strain. Equation (18) is as follows:

$$\varepsilon_V(x,y,t) = Max\left[\varepsilon_V^0(x,y,t) - f\varphi(x,y,t), 0\right] \tag{18}$$

where $f$ is the filling fraction. The capillary porosity, $\varphi(x, y, t)$, is dependent on the initial porosity before sulphate exposure, $\varphi_0$, and also the part filled by the resulting ettringite, $\alpha_s C_r(x, y, t)$. Accordingly, the instant value for $\varphi(x, y, t)$ may be updated per Equation (19), as follows:

$$\varphi(x,y,t) = Max[\varphi_0 - \alpha_S C_r(x,y,t), 0] \tag{19}$$

The volumetric strain is further transformed into the linear strain, based on the isotropy assumption in material mechanics. In this manner, the magnitude of volumetric strain is assumed to be the superposition of the linear strain along three principal directions [37]. Thus, the average linear strain corresponds to one-third of the volumetric strain, as shown in Equation (20).

The diffusivity of cementitious materials will be magnified, particularly after the sulphate-induced cracking occurs. Therefore, a diffusivity multiplier, $M_D$, should be introduced to take such an effect into account, as shown in Equations (21)–(23). According to Sarkar et al. [38], the effect of cracking on the ionic diffusion may be expressed as a function of crack density, $C_d$, which is in turn related to the instantaneous tensile strain, $\varepsilon$, and the tensile resistance of materials, $\varepsilon_{tp}$. Here, $k$ and $m_c$ are the tuning parameters which are recommended as 0.16 and 2.3, respectively [38]. Furthermore, $C_{dc}$ and $C_{dec}$ are two empirical percolation thresholds, and their values are suggested as 0.18 and 0.56, respectively [38]. Equations (20)–(23) are as follows:

$$\varepsilon(x,y,t) = \frac{\varepsilon_1 + \varepsilon_2 + \varepsilon_3}{3} = \frac{\varepsilon_V(x,y,t)}{3} \tag{20}$$

$$M_D = \left(1 + \frac{32}{9}C_d\right) + D_p \tag{21}$$

$$C_d = \begin{cases} 0 & \text{when } \varepsilon < \varepsilon_{tp} \\ k\left(1 - \frac{\varepsilon_{tp}}{\varepsilon}\right)^{m_C} & \text{when } \varepsilon \geq \varepsilon_{tp} \end{cases} \tag{22}$$

$$D_p = \begin{cases} 0 & \text{when } C_d \leq C_{dc} \\ (C_d - C_{dc})^2/(C_{dec} - C_d)^2 & \text{when } C_{dc} < C_d < C_{dec} \\ (C_{dec} - C_{dc})^2 & \text{when } C_d \geq C_{dec} \end{cases} \tag{23}$$

### 2.5. Durability-Based Limit State Function

For a given depth, the cracking time is defined as the moment at which the resulting strain derived from Equation (20) equals the maximum tensile capacity of concrete. Understandably, the ettringite and the associated expansion fill up the capillary pores along a certain direction. This sulphate-induced cracking manifests in a reduction in hardened properties on the one hand and an acceleration in the subsequent ionic transport on the other hand. More importantly, the intact region becomes vulnerable to external deleterious chemicals. Thus, a durability-based limit state function is proposed first, in the form of Equation (24). As seen therein, it mathematically reflects a difference between the designed cover thickness, $c$, and the cracked depth, $d_c(t)$. Once the latter becomes greater than the former, the value obtained from Equation (24) turns negative, and the corresponding time stamp is accordingly recognized as the failure of concrete structures. Equation (24) is as follows:

$$\text{g}(c, d_c(t)) = c - d_c(t) \tag{24}$$

If the thickness of the clear cover is known, the lifetime of concrete could be assessed by monitoring the instant strain at the clear cover, i.e., $\varepsilon(c, c, t)$. Combining Equations (13)–(20), the instant strain at the end of the clear cover is updated as the following Equation (25):

$$\varepsilon(c,c,t) = \left( \left( 1 - \frac{C(c,c,t)}{C_0} \right) \left( \frac{\Delta V}{V} \right)_{CA} - f\varphi(c,c,t) \right) \times \frac{1}{3} \qquad (25)$$

Together with the tensile capacity of concrete, i.e., $\varepsilon_{tp}$, the durability-based limit state function is eventually expressed in Equation (26) to evaluate the failure of concrete structures when subjected to the external sulphate attack. Equation (26) is as follows:

$$Z_{LSF} = \varepsilon_{tp} - \left( \left( 1 - \frac{C(c,c,t)}{C_0} \right) \left( \frac{\Delta V}{V} \right)_{CA} - f\varphi(c,c,t) \right) \times \frac{1}{3} \qquad (26)$$

## 3. Results and Discussion

### 3.1. Validation of Sulphate Penetration Based on Three-Phase Diffusion–Reaction Model

The proposed three-phase model is validated by comparing the numerical results with the experimental data generated in related studies [12,17]. The mortar sample examined in the first report was sized as 50 mm × 50 mm × 15 mm, with an embedded coarse aggregate [12]. The concrete specimen produced in the second report was 100 mm × 100 mm × 100 mm in dimension [17] and was made from a blended Portland cement incorporating 30% fly ash. The chemical compositions constituting both binders are now listed in Table 1, and the associated mixing proportions are shown in Table 2. In those two studies, the specimens were firstly aged for 28 days, and were then immersed in the sulphate-rich solution, containing 5% anhydrous sodium sulphate (equivalent to 352 mol/m$^3$ in molar concentration). After 84 days [12] and 300 days [17] of exposure, respectively, the specimens were variously retrieved to measure the sulphate concentration via chemical titration. In the meantime, the numerical profiles generated from the proposed three-phase model are compared against those two sets of experimental data in Figure 1. It is clear that the predicted sulphate ingress captures the actual measurements. This implies that the proposed three-phase diffusion–reaction model is efficient to simulate the sulphate diffusion–reaction behavior in concrete when exposed to a sulphate-rich environment.

**Table 1.** Chief chemical compositions of Portland cement and fly ash.

| Oxide | CaO | SiO$_2$ | Al$_2$O$_3$ | Fe$_2$O$_3$ | SO$_3$ | MgO | K$_2$O | R$_2$O | Na$_2$O | TiO$_2$ |
|---|---|---|---|---|---|---|---|---|---|---|
| Cement [12] | 62.7% | 20.4% | 4.6% | 3.4% | 2.7% | 2.8% | 0.5% | - | - | - |
| Cement [17] | 58.1% | 22.1% | 6.7% | 4.43% | 2.22% | 0.87% | 0.37% | 0.54% | 0.3% | - |
| Fly ash | 11.0% | 55.5% | 23.2% | 3.6% | 0.2% | 1.2% | 0.8% | - | 2.8% | 0.7% |

**Table 2.** Mix proportions for various concrete systems.

| Mixture | W/B | Fly Ash | Water | Cement | Fine Aggregate | Coarse Aggregate | Admixture |
|---|---|---|---|---|---|---|---|
| Reference [12] | 0.485 | 160.5 kg | 260 kg | 374.5 kg | 1470 kg | - | - |
| Reference [17] | 0.45 | 80 kg | 180 kg | 320 kg | 710 kg | 1060 kg | 0.383% |

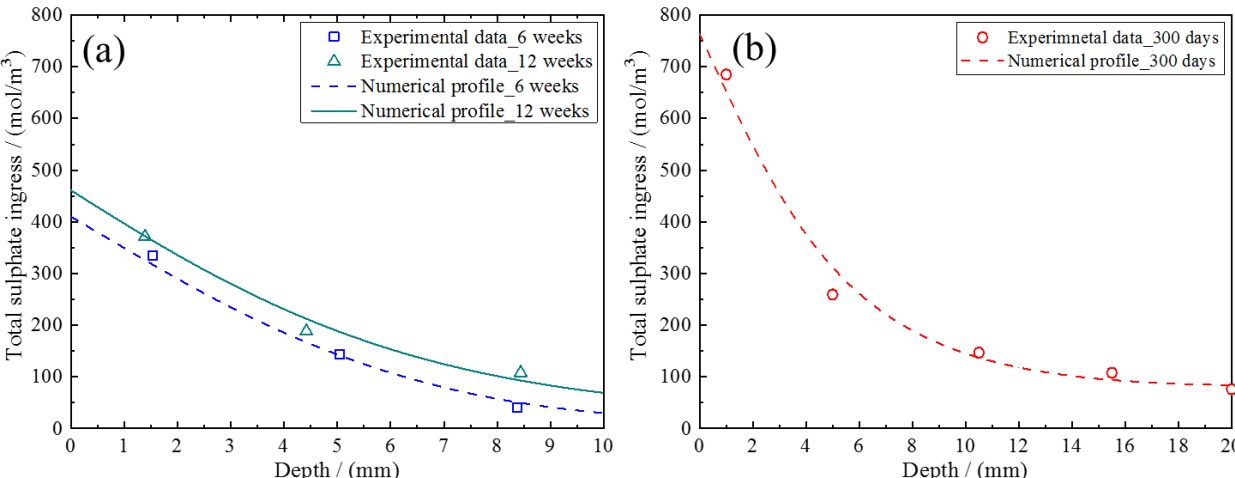

**Figure 1.** (**a**,**b**) Comparing the predicted sulphate ingress with experimental data (Adapted from the authors' prior studies [12,17]).

### 3.2. Parametric Analysis

In the present study, a parametric analysis is carried out to assess the effect of each phase on sulphate attack. The diffusion coefficients of mortar, coarse aggregate, and ITZ, alongside their volumetric fractions, are variously examined. All the parameters, except the examined factors, were constantly kept invariant throughout this parametric analysis. These invariants are as follows: $U_0 = 5\%$ (352 mol/m$^3$), $C_0 = 10\%$ by weight of binder, $k_0 = 1.0 \times 10^{-7}$ mol/m$^3$·s, $\varphi_0 = 0.08$, $f = 0.10$, $\lambda = 2.4535$, and $\varepsilon_t = 1.5 \times 10^{-4}$ [12]. The profiles obtained at 20 years are then presented. Note here that the parametric analysis is mainly conducted based on the three-phase diffusion–reaction model which has currently been validated for sulphate penetration only. In the future work, further validations are required from the aspects of sulphate-induced strain and even the crack propagation.

#### 3.2.1. Diffusivity of Mortar

The numerical investigation accounted for four different levels of the initial diffusion coefficient for mortar. Here, $D_{p0}$ was variously set as $1.0 \times 10^{-12}$ m$^2$/s, $3.0 \times 10^{-12}$ m$^2$/s, $5.0 \times 10^{-12}$ m$^2$/s, and $7.0 \times 10^{-12}$ m$^2$/s. Apart from the parameters mentioned at the outset, the diffusion coefficients of coarse aggregates and ITZs alongside their fractions were fixed as a constant in this case, namely $D_a = 0.001 D_{p0}$, $D_{ITZ} = 16.2 D_{p0}$, $f_a = 0.35$ and $f_{ITZ} = 0.001$. Note here that besides sulphate concentration and ettringite production, the resulting tensile strain is also presented here, as it is directly related to the crack propagation induced by sulphate attack. The corresponding numerical profiles against varying depth are now plotted in Figure 2. Clearly, one sees that the diffusivity of the mortar affects the sulphate attack substantially, as evident from the significant rise in sulphate concentration and ettringite production, and the resulting strain with an increase in $D_{p0}$. As is well known, Portland cement concrete is a porous material, and the mortar is the dominant phase in such a three-phase system. Accordingly, the greater diffusivity of the mortar indicates the faster transport of sulphate ions inside concrete. This promotes the accumulation of sulphate ions and, later, brings more reactants to react with the inherent calcium aluminate. Eventually, the ettringite formation and the ensuing sulphate-induced strain are also boosted. Note further from Figure 2b,c that, regardless of the value of $D_{p0}$, the predicted profiles at the shallow location converge at the same level. This implies that the overall amount of sulphate-induced distress may be independent upon the diffusivity of the mortar. Nevertheless, it should be emphasized that an increase in the diffusion coefficient of mortar will significantly shorten the period required to reach the maximum sulphate-induced strain.

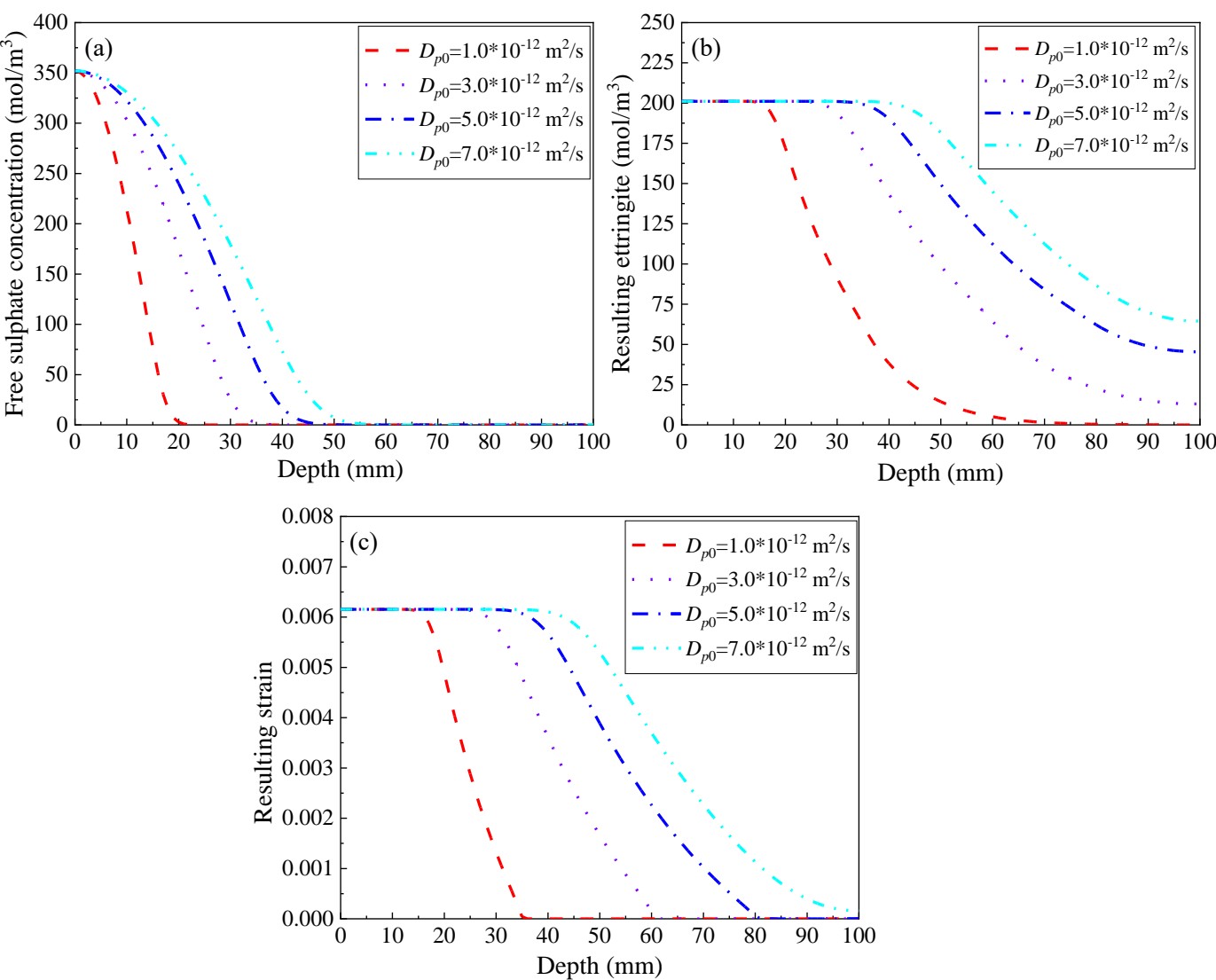

**Figure 2.** Effect of diffusivity of mortar on (**a**) sulphate concentration, (**b**) ettringite production, and (**c**) resulting strain.

### 3.2.2. Diffusivity of Coarse Aggregate

Coarse aggregate takes up the second largest phase in the hardened concrete system. However, compared to the other two phases (mortar and ITZ), the diffusivity of coarse aggregate is substantially smaller. Hence, the penetrating sulphate ions diffuse very slowly in this phase, and the role of coarse aggregate may act as a "deceleration strip" on the path of sulphate diffusion. To quantify the influence of coarse aggregates, the corresponding diffusion coefficient was set from 0.1 to 0.001 times the value of mortar, $D_{p0}$. Again, other parameters are kept constant, as follows: $D_{p0} = 2.26 \times 10^{-12}$ m$^2$/s, $D_{ITZ} = 16.2\, D_{p0}$, $f_a = 0.35$, and $f_{ITZ} = 0.001$. Figure 3 reveals sulphate concentration, ettringite production, and the resulting tensile strain under the external sulphate attack. As seen therein, the influence due to varying diffusivity of coarse aggregate may be distinguished into two stages. Firstly, when $D_a$ drops from 0.1 $D_{p0}$ to 0.01 $D_{p0}$, the progress of sulphate attack is obviously deterred, manifesting in a noticeable reduction in sulphate concentration, ettringite formation, and the resulting strain. This is strongly connected to the hindrance led by the coarse aggregate. In this regard, the smaller the diffusion coefficient, the stronger the hindrance upon sulphate penetration and, accordingly, the less sulphate-induced distress. However, as the value of $D_a$ drops down to a certain level, here noted as 0.01 $D_{p0}$, any further decrease in this parameter will no longer obviously alleviate the external

sulphate attack. This is likely attributed to the fact that the embedded coarse aggregate with a diffusivity of 0.1~0.01 $D_{p0}$ could still be recognized as a diffusible phase, but the aggregate phase with a $D_a$ below 0.01 $D_{p0}$ may be treated as a non-diffusible state. The coarse aggregate in the former case only acts as a "deceleration strip", whereas this phase appears to be a "dam" in the latter case. Given the above, controlling the permeability of coarse aggregates will help to deter the penetration of sulphate ions. More importantly, employing coarse aggregates with a diffusion coefficient of 0.01 $D_{p0}$ may maximize the obstructing effect led by coarse aggregates on sulphate attack in concrete.

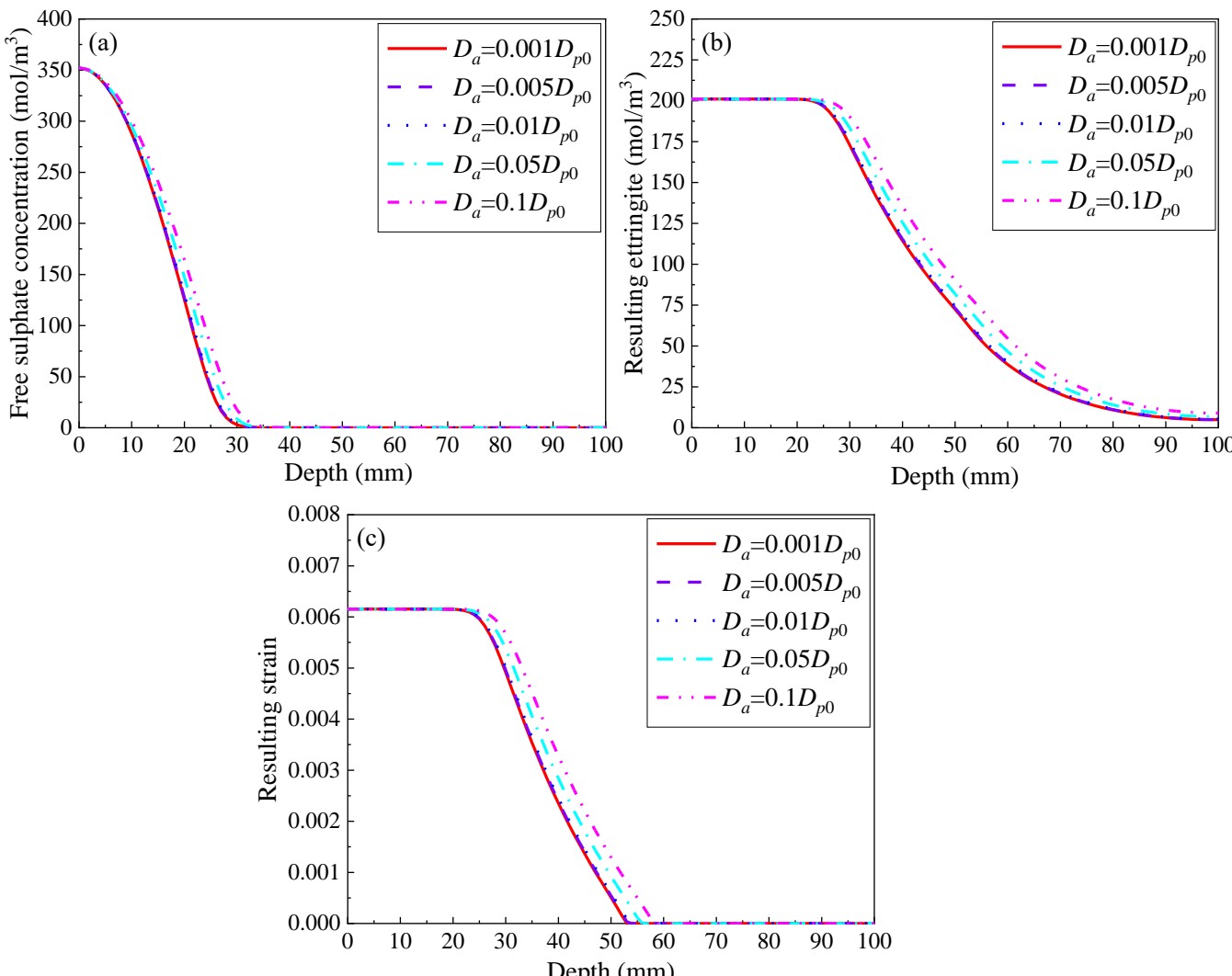

**Figure 3.** Effect of diffusivity of coarse aggregate on (**a**) sulphate concentration, (**b**) ettringite production, and (**c**) resulting strain.

### 3.2.3. Diffusivity of ITZ

The ITZ is a porous region between the paste matrix and the coarse aggregates. Naturally, this region registers a local increase in micro-cracks compared to the other two phases. Accordingly, the ITZ is widely recognized as the weakest phase in the hardened concrete [39]. In this study, it is assumed that the ITZ exists around the surface of coarse aggregates with an identical thickness. Prior studies report that its diffusivity is about 1.3 to 16.2 times that of mortar [29]. Hence, four diffusion coefficients, namely $D_{ITZ} = 1.2D_{p0}$, $D_{ITZ} = 6.2D_{p0}$, $D_{ITZ} = 11.2D_{p0}$, and $D_{ITZ} = 16.2D_{p0}$, were examined in the present study to investigate the diffusivity of ITZ on sulphate attack in concrete. Likewise, other factors are set as invariants, as follows: $D_{p0} = 2.26 \times 10^{-12}$ m²/s, $D_a = 0.001D_{p0}$, $f_a = 0.35$, and

$f_{ITZ}$ = 0.001. The sulphate concentration, ettringite production, and the resulting tensile strain predicted from the three-phase model are now plotted in Figure 4. Interestingly, these outputs are not sensitive to the diffusion coefficient of ITZ, as is evident from the overlapped profiles shown in Figure 4. Furthermore, a very minor difference is found even after zooming into Figure 5 at the depth of 24.5~25 mm. The principal reason behind this may be that the fraction of ITZ in concrete is significantly small in comparison to the mortar or the coarse aggregates. According to a related study, the thickness of ITZ typically varies from 5 μm to 65 μm [29]. This means that the corresponding volumetric fraction only takes up about 1% or even lower of the total mixture. Given this, while the diffusivity of ITZ is larger than the other two phases, it does not pose a substantial influence on sulphate attack in concrete.

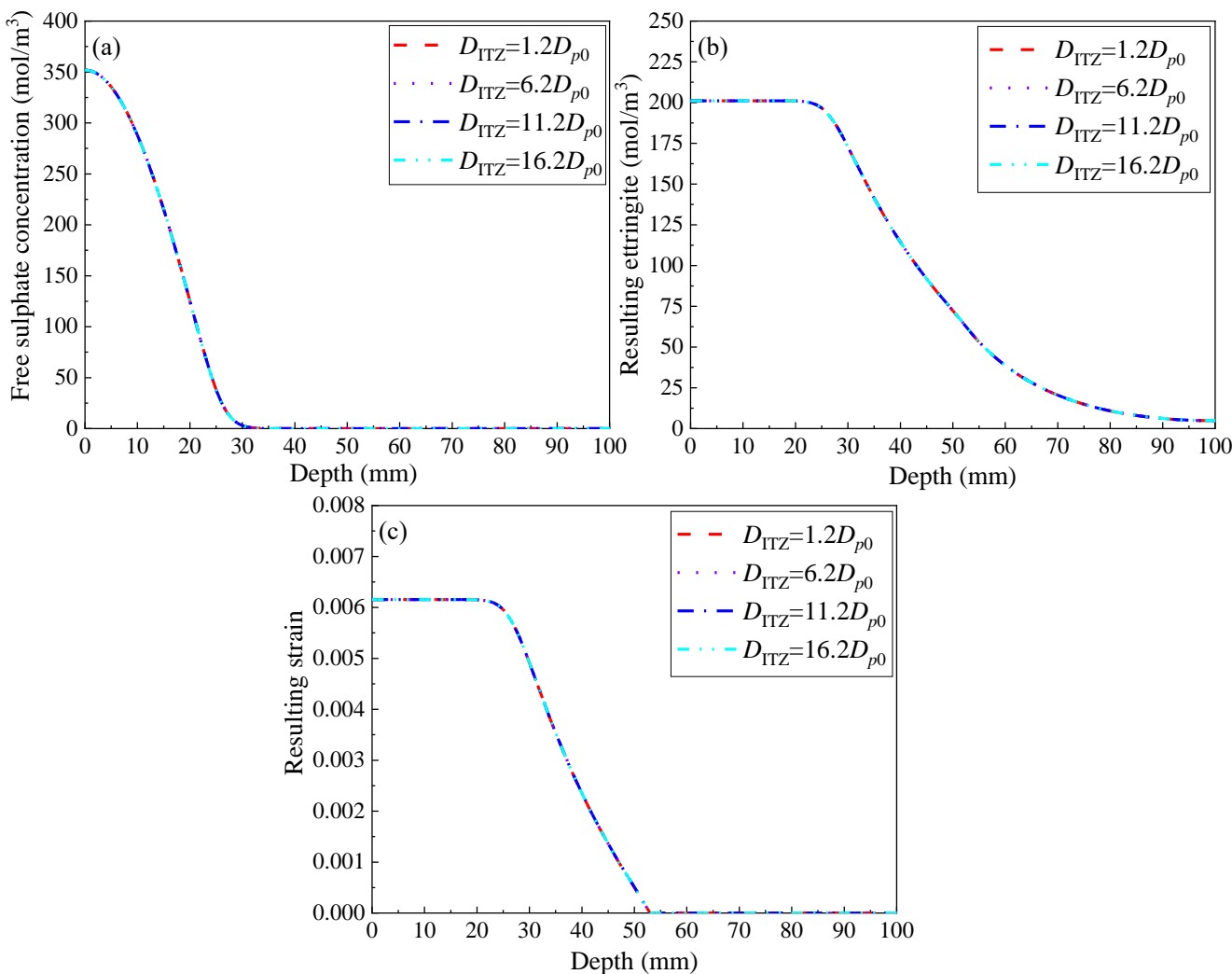

**Figure 4.** Effect of diffusivity of ITZ on (**a**) sulphate concentration, (**b**) ettringite production, and (**c**) resulting strain.

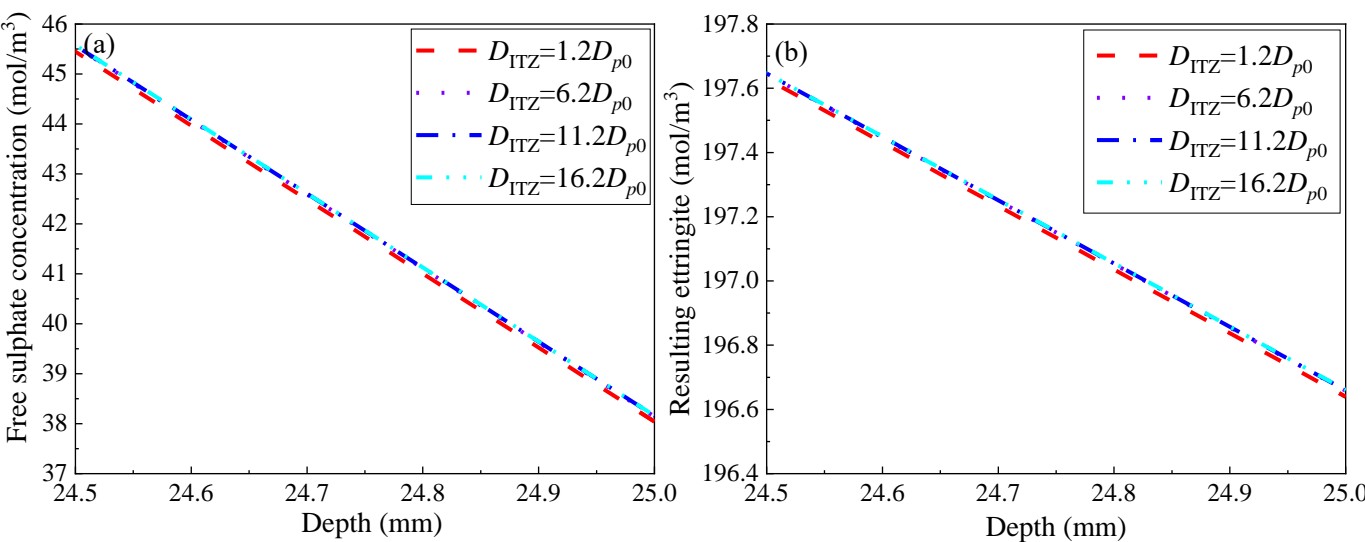

**Figure 5.** *Cont.*

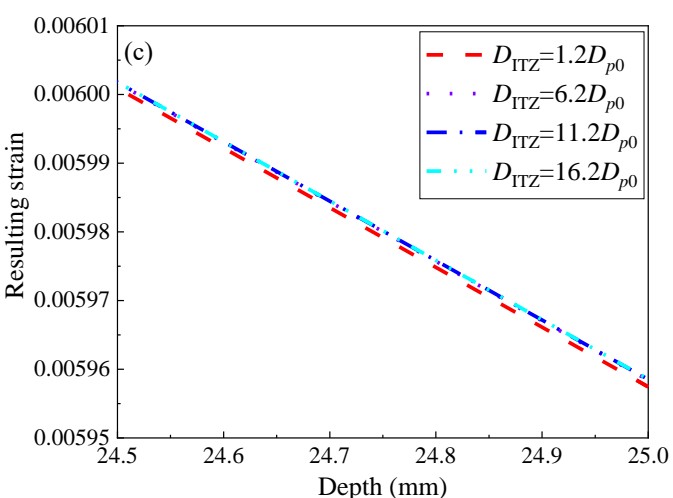

**Figure 5.** The zoomed-in profiles of (**a**) sulphate concentration, (**b**) ettringite production, and (**c**) resulting strain at 24.5~25 mm in Figure 4.

### 3.2.4. Volumetric Fraction of Coarse Aggregate

Besides the diffusivity, the volumetric fraction of each phase also determines the ability of chemical ions to migrate inside concrete. In order to measure the effect of aggregate fraction, $f_a$, on sulphate-induced distress, the value of $f_a$ is variously adjusted from 0.20 to 0.50 with an increment of 0.05. Once again, other parameters are fixed as follows: $D_{p0} = 2.26 \times 10^{-12}$ m$^2$/s, $D_{ITZ} = 16.2 D_{p0}$, $D_a = 0.001 D_{p0}$, and $f_{ITZ} = 0.001$. It is noted from Figure 6 that the sulphate attack is indeed sensitive to the fraction of coarse aggregates. Clearly, the hardened concrete made with the larger $f_a$ will experience much less sulphate-induced deterioration, manifesting in the lower sulphate ingress and lower ettringite formation alongside a smaller tensile strain. Each individual coarse aggregate acts as a "deceleration strip", which effectively slows down the penetration of sulphate ions due to the smaller diffusion coefficient. Furthermore, due to the extremely low calcium content, coarse aggregates are able to reduce the global amount of ettringite under sulphate attack. This in turn alleviates the scale of tensile strain. Furthermore, owing to the significantly low diffusivity of coarse aggregates, the tortuosity of sulphate diffusion in the mortar and ITZs is raised, and this evolves with an increase in the $f_a$. Taken together, it is not surprising

to note that the modelled concrete that contains a greater fraction of coarse aggregates displays a lower sulphate-induced distress.

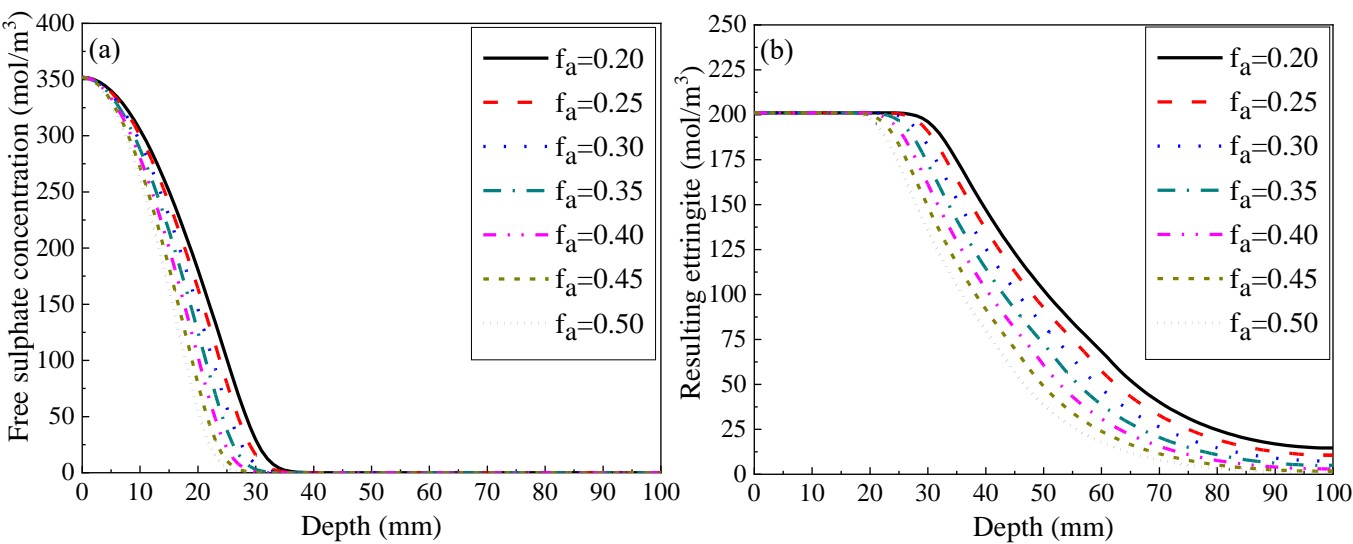

**Figure 6.** *Cont.*

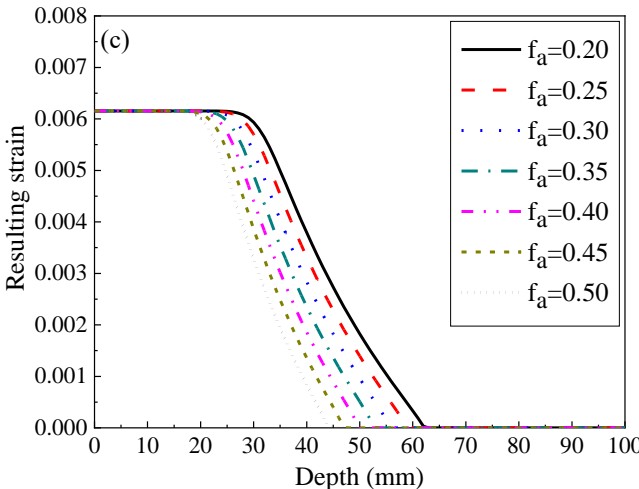

**Figure 6.** Effect of coarse aggregate fraction on (**a**) sulphate concentration, (**b**) ettringite production, and (**c**) resulting strain.

### 3.2.5. Volumetric Fraction of ITZ

The ITZ registers the largest diffusion coefficient across all three phases, and such a region performs as an "acceleration zone" when subjected to the penetration of sulphate ions. Recall that the thickness of ITZ was reported to be about 5~65 μm [29]. It is assumed here that ITZ exists around the surface of spherical coarse aggregates. As such, the above thickness may approximately correspond to a volume fraction of 1.0~0.05%, with a coarse aggregate radius of 10 mm and an aggregate fraction of 0.35~0.5. Accordingly, the sensitivity of sulphate-induced distress to the fraction of ITZ, $f_{ITZ}$, is numerically investigated within 0.0005~0.01 (0.05~1%). In the meantime, other parameters are set as constant, as before. Now, the sulphate concentration, ettringite production, and the resulting tensile strain are presented in Figure 7. Clearly, the variation in $f_{ITZ}$ does not cause substantial change in these outputs. Nevertheless, when we magnify these profiles at a certain depth, e.g., 24.5~25 mm, the difference is indeed detected. Seen from Figure 8, an increase in the fraction of ITZ corresponds to a slight increase in all three outputs. This implies that while the fraction of

ITZ is significantly smaller than mortar and coarse aggregate, increasing this variable may slightly promote the progress of sulphate attack in concrete. A similar phenomenon was noted in another related study in terms of chloride attack, in which a minor difference was also reported with an increase in the size of ITZ [29].

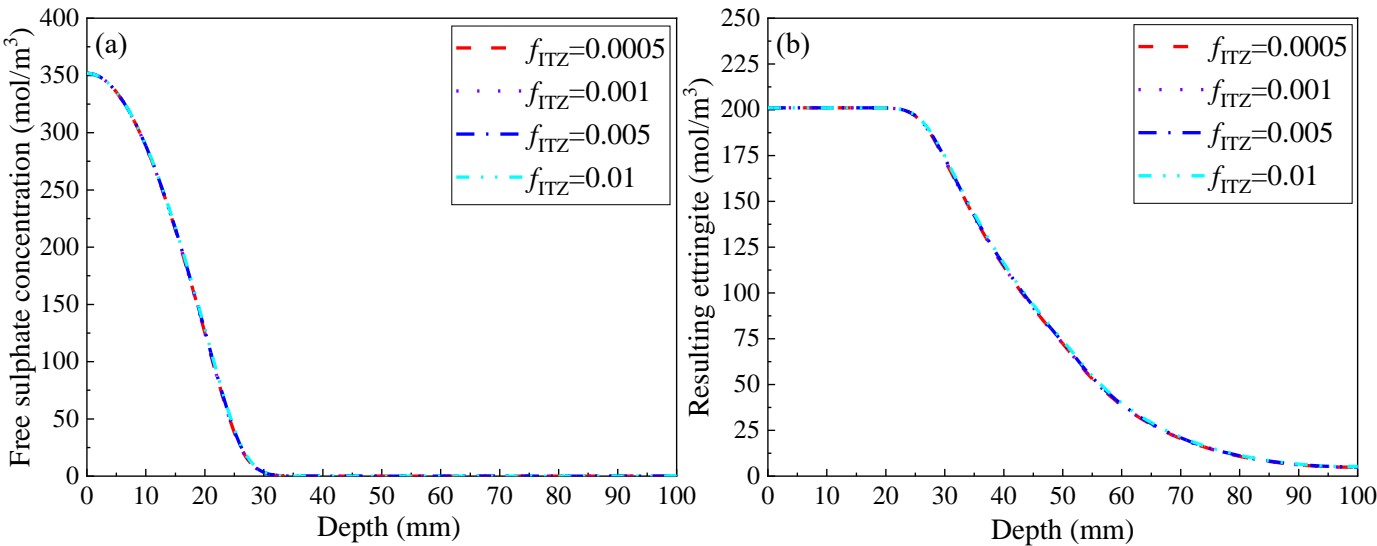

**Figure 7.** *Cont.*

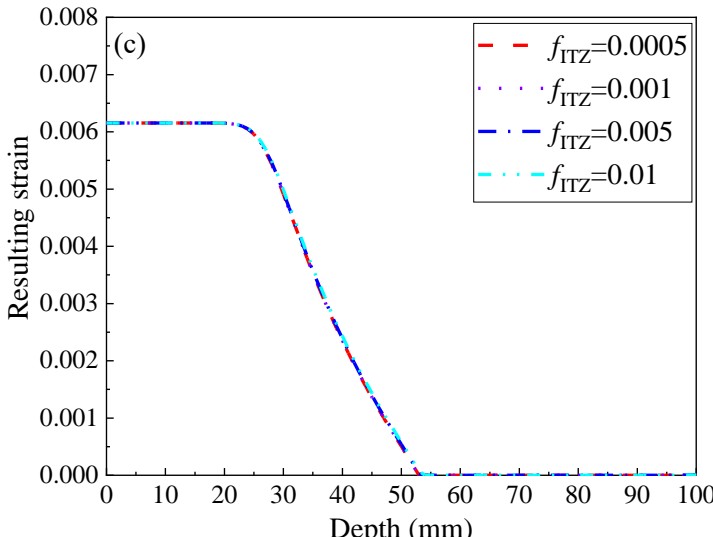

**Figure 7.** Effect of ITZ fraction on (**a**) sulphate concentration, (**b**) ettringite production, and (**c**) resulting strain.

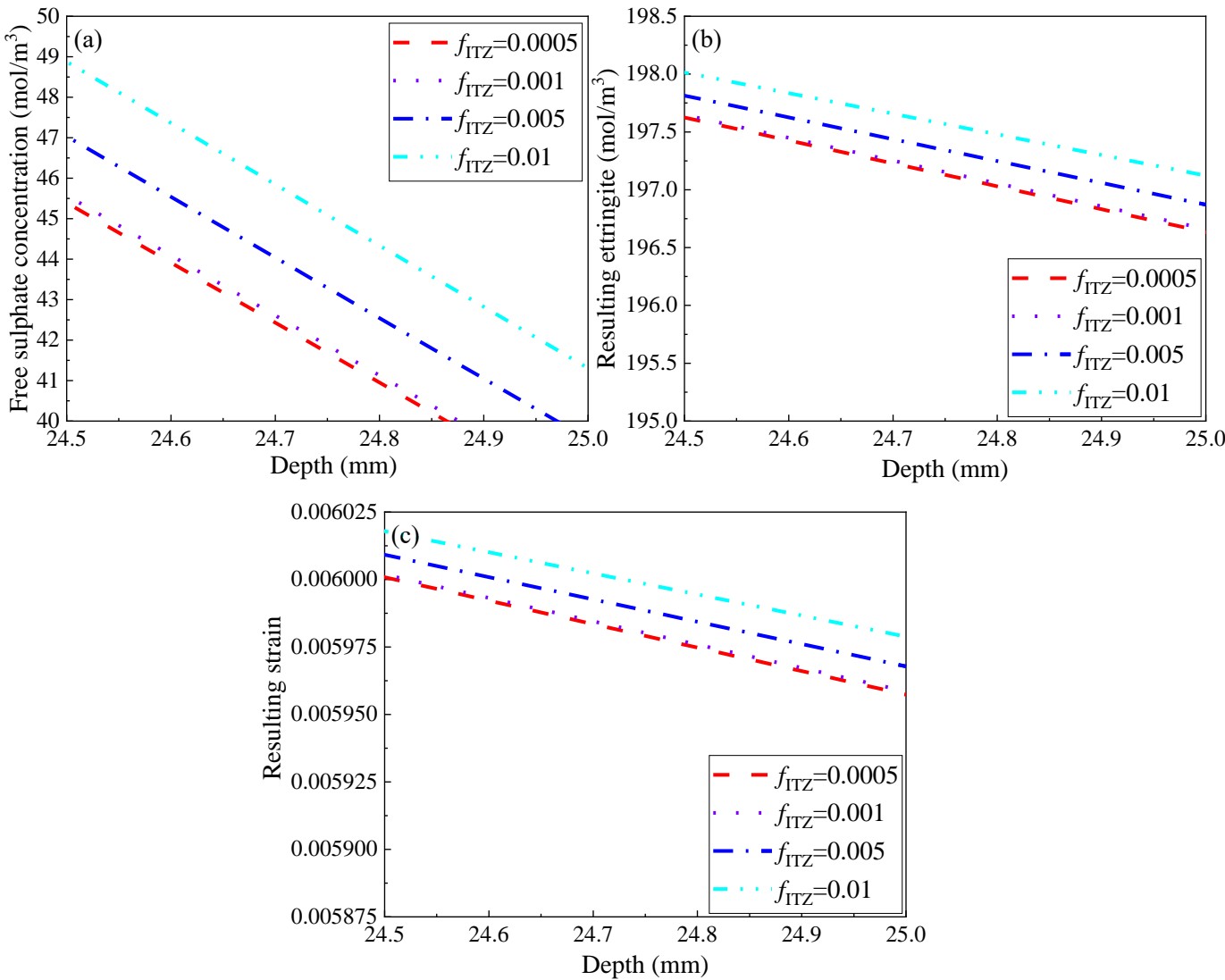

**Figure 8.** The zoomed-in profiles of (**a**) sulphate concentration, (**b**) ettringite production, and (**c**) resulting strain at 24.5~25 mm in Figure 7.

### 3.3. Crack Propagation Prediction and Sensitivity Analysis

The crack propagation in concrete structures exposed to sulphate attack may be predicted as per Equation (26). Recall that the failure is defined as the time instant at which the sulphate-induced crack growth penetrates through the entire cover thickness of the concrete. Note here that, besides the resulting strain, the output of the proposed durability state function is also dependent upon the tensile capacity of concrete, as well as on the designed cover thickness. In a related study, the tensile resistance of concrete, $\varepsilon_t$, was experimentally measured as $1.5 \times 10^{-4}$~$2.0 \times 10^{-4}$ [12]. Furthermore, the minimum value of clear cover is recommended as 1.5–2.0 times the maximum aggregate size (assuming 20 mm in the present study) for concrete members to resist sulphate and/or chloride attacks, as per CAN/CSA-A23.1 [40]. This corresponds to a cover thickness set, respectively, as 30 mm and 40 mm.

Based on the predicted cracking time, a variance-based sensitivity analysis was carried out. The underlying principle relates the total variance of predicted values to the corresponding partial variance due to the variation in each factor [41]. Here, let $y(x)$ define the predicted cracking time with various examined factors for $x = \{x_1, x_2, \ldots, x_l\}$. In the conventional statistical analysis, the resulting variance usually fails to capture the effect of the heterogeneous distribution of factors. Hence, an increment coefficient $\alpha_{km}$ is introduced

here to normalize the variance, yielding the effective partial variance, $V_e$. Eventually, the sensitivity index, $S$, for each factor is obtained by dividing the effective partial standard variance by the total effective standard variance. The above statistical analysis is now mathematically expressed in the following Equations (27)–(30):

$$V_{te} = \sum_{i=1}^{n} V_{ei} \tag{27}$$

$$V_{ei} = \frac{1}{N} \sum_{k=1}^{N} \frac{(y_{ik} - y_{im})^2}{\alpha_{km}} \tag{28}$$

$$\alpha_{km} = \frac{x_{ik} - x_{im}}{x_{im}} \tag{29}$$

$$S_i = \sqrt{\frac{V_{ei}}{V_{te}}} \in [0, 1] \tag{30}$$

Here, the number of examined factors and their dimensions are respectively denoted as $n$ and $N$, while $m$ defines the average value of each factor.

Figure 9 presents the time-to-cracking at the clear cover against varying (a) $D_{p0}$, (b) $D_a$, (c) $D_{ITZ}$, (d) $f_a$, and (e) $f_{ITZ}$. Note that regardless of the cover thickness, the diffusivity of mortar plays the most significant role across all five parameters. This is followed by the volumetric fraction and the diffusivity of coarse aggregate, respectively. On the other hand, any variation on the diffusivity or/and the volumetric fraction of ITZ generates a negligible influence on the sulphate-induced distress. Note here that the cracking instance results are well-supported by the former parametric analysis, as the tendency respectively coincides with that seen on profiles of sulphate concentration, ettringite production, and tensile strain in Section 3.2. Prior to determining the sensitivity index, the average value for each examined factor, i.e., $x_{im}$, and the associated output, $y_{im}$, are needed, as now shown in Figure 10. The calculating details involved in this sensitivity analysis are presented in Table 3, and Figure 11 reveals the sensitivity indices of examined factors. Clearly, when subjected to a sulphate-rich environment, the expansive cracking in concrete structures is most sensitive to the diffusivity of mortar ($D_{p0}$), as is evident from the highest sensitivity index of 73.645%. The volumetric fraction ($f_a$) and diffusivity ($D_a$) of coarse aggregate have a lower impact, ranking second ($S_i = 20.948\%$) and third ($S_i = 4.899\%$), respectively. On the other hand, the diffusion coefficient and fraction of ITZ are found to be least significant across all five examined factors, representing no more than 0.6% in total.

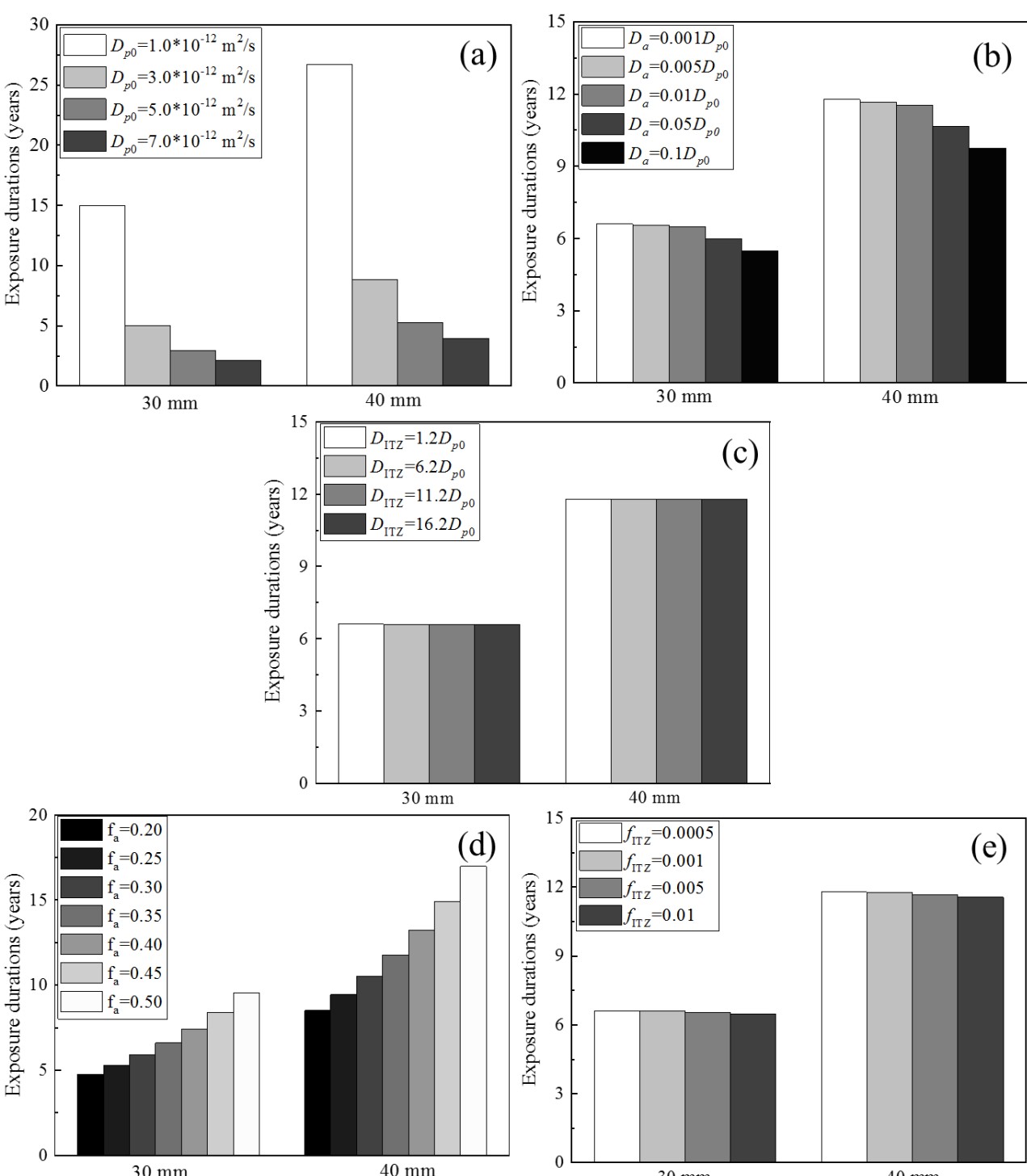

**Figure 9.** Predicted cracking instance for concrete systems with varying (**a**) mortar diffusivity, $D_{p0}$, (**b**) coarse aggregate diffusivity, $D_a$, (**c**) ITZ diffusivity, $D_{ITZ}$, (**d**) coarse aggregate fraction, $f_a$, and (**e**) ITZ fraction, $f_{ITZ}$.

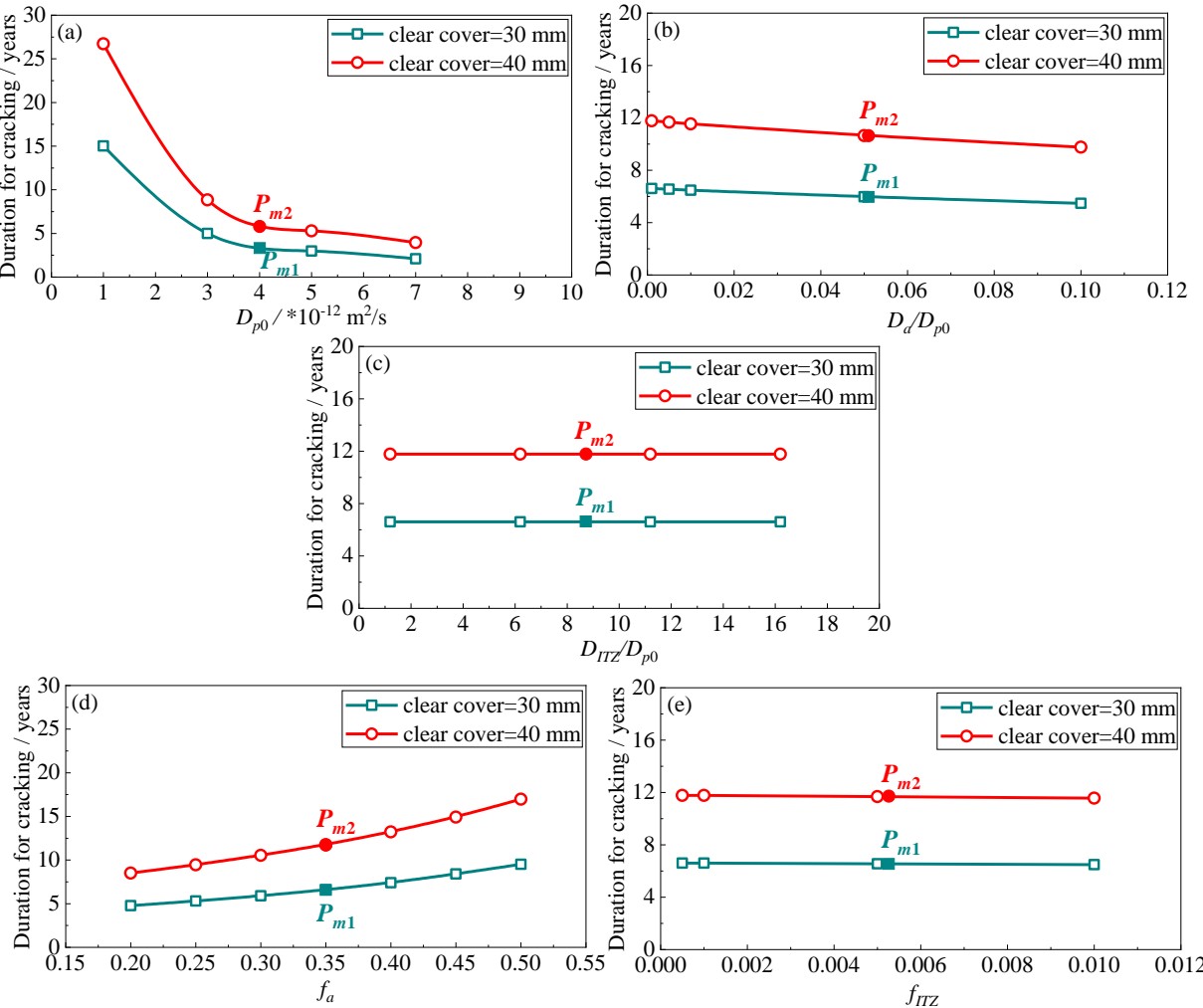

**Figure 10.** Average value for each examined factor in sensitivity analysis. (**a**) Mortar diffusivity, $D_{p0}$, (**b**) coarse aggregate diffusivity, $D_a$, (**c**) ITZ diffusivity, $D_{ITZ}$, (**d**) coarse aggregate fraction, $f_a$, and (**e**) ITZ fraction, $f_{ITZ}$.

**Table 3.** Sensitivities of cracking instance from $D_{p0}$, $D_a$, $D_{ITZ}$, $f_a$, and $f_{ITZ}$.

| Factor | $x_{ik}$ | $y_{ik}$ (30 mm) | $y_{ik}$ (40 mm) | $x_{im}$ | $y_{im}$ (30 mm) | $y_{im}$ (40 mm) | $\alpha_{km}$ | $S_{i1}$ (%) (30 mm) | $S_{i2}$ (%) (40 mm) | $S_i$ (%) |
|---|---|---|---|---|---|---|---|---|---|---|
| $D_{p0}$ | 1 | 15.01 | 26.72 | 4 | 3.40 | 5.80 | 0.75 | 73.351 | 73.939 | 73.645 |
|  | 3 | 4.99 | 8.85 |  |  |  | 0.25 |  |  |  |
|  | 5 | 2.99 | 5.28 |  |  |  | 0.25 |  |  |  |
|  | 7 | 2.10 | 3.95 |  |  |  | 0.75 |  |  |  |
| $D_a$ | 0.001 | 6.60 | 11.77 | 0.0505 | 5.965 | 10.64 | 0.980198 | 4.956 | 4.843 | 4.899 |
|  | 0.005 | 6.55 | 11.67 |  |  |  | 0.90099 |  |  |  |
|  | 0.01 | 6.47 | 11.54 |  |  |  | 0.80198 |  |  |  |
|  | 0.05 | 5.97 | 10.65 |  |  |  | 0.009901 |  |  |  |
|  | 0.1 | 5.47 | 9.76 |  |  |  | 0.980198 |  |  |  |
| $D_{ITZ}$ | 1.2 | 6.61 | 11.78 | 8.7 | 6.60 | 11.77 | 0.862069 | 0.057 | 0.031 | 0.044 |
|  | 6.2 | 6.60 | 11.77 |  |  |  | 0.287356 |  |  |  |
|  | 11.2 | 6.60 | 11.77 |  |  |  | 0.287356 |  |  |  |
|  | 16.2 | 6.60 | 11.77 |  |  |  | 0.862069 |  |  |  |

**Table 3.** *Cont.*

| Factor | $x_{ik}$ | $y_{ik}$ (30 mm) | $y_{ik}$ (40 mm) | $x_{im}$ | $y_{im}$ (30 mm) | $y_{im}$ (40 mm) | $\alpha_{km}$ | $S_{i1}$ (%) (30 mm) | $S_{i2}$ (%) (40 mm) | $S_i$ (%) |
|---|---|---|---|---|---|---|---|---|---|---|
| $f_a$ | 0.20 | 4.77 | 8.51 | 0.35 | 6.60 | 11.77 | 0.428571429 | 21.184 | 20.711 | 20.948 |
| | 0.25 | 5.30 | 9.45 | | | | 0.285714286 | | | |
| | 0.30 | 5.91 | 10.53 | | | | 0.142857143 | | | |
| | 0.40 | 7.41 | 13.22 | | | | 0.142857143 | | | |
| | 0.45 | 8.40 | 14.93 | | | | 0.285714286 | | | |
| | 0.50 | 9.52 | 16.98 | | | | 0.428571429 | | | |
| $f_{ITZ}$ | 0.0005 | 6.61 | 11.78 | 0.00525 | 6.547 | 11.674 | 0.904761905 | 0.452 | 0.476 | 0.464 |
| | 0.001 | 6.60 | 11.77 | | | | 0.80952381 | | | |
| | 0.005 | 6.55 | 11.68 | | | | 0.047619048 | | | |
| | 0.01 | 6.49 | 11.56 | | | | 0.904761905 | | | |

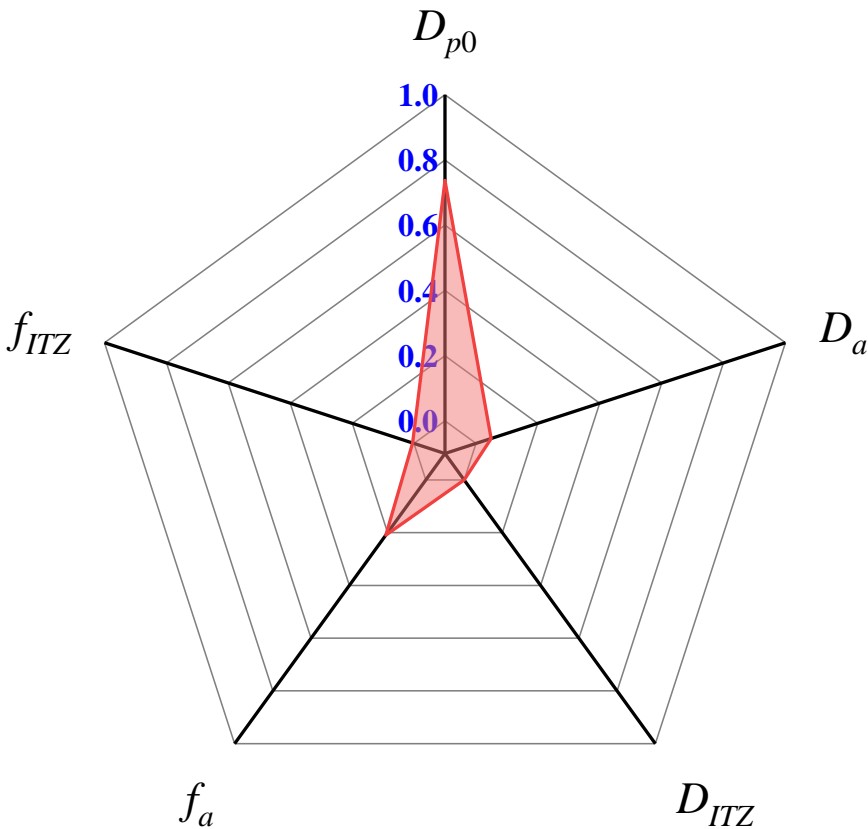

**Figure 11.** Sensitivities of sulphate-induced cracking to $D_{p0}$, $D_a$, $D_{ITZ}$, $f_a$, and $f_{ITZ}$.

Based on these numerical simulations, as well as the sensitivity analysis, the present authors recognize that any approach that is able to reduce the diffusivity of the cement matrix is believed to enhance the resistance of concrete to external sulphate attack. Furthermore, employing coarse aggregates with low permeability and increasing the aggregate fraction within a workable range should also help to alleviate the sulphate-induced distress in concrete.

## 4. Concluding Remarks

Concrete is widely considered as a homogeneous diffusion medium when simulating sulphate attack. The associated models, therefore, fail to capture the effect of each individual phase in this durability concern. In order to fill in the above research gap, the present study constructs concrete as a three-phase system to account for the diffusivity and volumetric

fraction of mortar, coarse aggregates, and ITZs, respectively. A three-phase diffusion–reaction model is, thereafter, proposed to simulate the external sulphate attack on concrete. Subsequently, together with the parametric analysis, an elaborate sensitivity analysis was carried out to quantify the significance of sulphate-induced cracking to the diffusivity alongside the volumetric fraction of each phase. Based on the obtained results, the following conclusions may be drawn:

1.  The sulphate penetration and the ensuing crack propagation are found to be most sensitive to the diffusivity of mortar. This is specifically evident from its highest sensitivity index in terms of the sulphate-induced cracking. In addition, the correlation between the diffusivity of mortar and the sulphate-induced cracking is not linear. When the diffusion coefficient of sulphate ions in mortar is below a certain value, here found as $1.0 \times 10^{-12}$ m$^2$/s, the sulphate attack progress may be deterred significantly;

2.  The lifetime of concrete structures is believed to extend with a decrease in the diffusivity of the coarse aggregate and with an increase in its volume fraction with respect to the entire mixture. This is due to its extremely low diffusion coefficient, as well as its negligible calcium aluminate content. Furthermore, increasing its fraction is a more promising method to improve the sulphate resistance of concrete;

3.  Although the ITZ is the weakest phase in the modelled concrete, varying its diffusivity and fraction may affect the progress of sulphate attack only very slightly. This is attributed to its extremely low volumetric fraction in comparison to that of mortar or coarse aggregates. Compared to the diffusivity of ITZ, the sulphate attack may be more sensitive to its fraction.

Requirement and recommendation for future work are as follows: The numerical analysis conducted in the present study is mainly based on the diffusion–reaction of sulphate ions and the prediction of sulphate-induced strain and cracking. The former has been validated through comparing the sulphate penetration profiles with actual experimental data. However, the validations for spatial strain led by sulphate attack and the ensuing crack propagation in concrete have not yet been sufficiently performed. Therefore, further validations for these two aspects are required in the future work.

**Author Contributions:** Conceptualization, C.Y.; methodology, C.Y., Z.C., J.Y. and V.B.; software, C.Y. and J.Y.; formal analysis, C.Y., Z.C. and J.Y.; investigation, C.Y.; writing—original draft, C.Y.; writing—review and editing, V.B.; supervision, Z.C. and V.B.; Project administration, Z.C. and V.B.; funding acquisition, Z.C. and V.B. All authors have read and agreed to the published version of the manuscript.

**Funding:** This research project was funded by Natural Sciences and Engineering Research Council (NSERC) Canada (RGPIN-2017-06870), the National Natural Science Foundation of China (NSFC U2006224) and the Guangxi Special Project for Innovation-driven Development (GKAA18242007 and GKAA18118029).

**Data Availability Statement:** The data that support the findings of this study are available from the authors upon reasonable request.

**Acknowledgments:** This study was a jointly conducted project between Guangxi University, China, and the University of Alberta, Canada. Chen is sponsored by the National Natural Science Foundation of China (NSFC U2006224) and the Guangxi Special Project for Innovation-driven Development (GKAA18242007 and GKAA18118029). Bindiganavile acknowledges the Natural Sciences and Engineering Research Council (NSERC) Canada for its continued financial support.

**Conflicts of Interest:** The authors declare no conflict of interest.

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
