# Peer review of "A Three-Phase Model to Evaluate Effects of Phase Diffusivity and Volume Fraction upon the Crack Propagation in Concrete Subjected to External Sulphate Attack"

_2673-4109, doi:10.3390/civileng4010002_

Round 1

Reviewer 1 Report

While the paper makes good reading, and is a nice contribution to the journal, the authors should make an attempt to improve the paper. Some comments are below:

1. Figure 1 is unnecessary.

2. 'Inputted' and 'Outputted' - Page 8 and also later: These words cannot be used as verb forms!

3. Lines 379 - 385 (page 13: It is stated that ITZ forms a small fraction and does not contribute to the attack (later also in conclusions). This statement needs justification. Please refer to classical papers on ITZ - the micrographical evidence shows that ITZ actually dominates the volume, since the distance between aggregates in a low to moderate strength concrete is less than 200 microns! Therefore, this assessment, as well as the conclusion listed in Section 4, are not really in line with the observations from several previous studies. The problem may lie in the assumption of the extent of ITZ (thickness around the aggregate) in this paper.

4. Lines 430 - 435: The overall conclusion from the work indicates something that is already well known!!

5. Apart from comparing the model data to just the sulphate penetration, the authors should also attempt to look at results from experiments where the rates of deterioration were clearly established.  Further, there is sufficient experimental evidence of the amount of ettringite forming in the mortar / concrete systems in several research articles. The authors should try to compare their model results with these studies too.

Author Response

The authors have revised our manuscript as per the reviewer's comments. Please find the attached authors' response file.

Reviewer 2 Report

This paper presents a numerical study focusing on the external sulfate attack of cementitious materials. The influence of phase compositions and service life prediction were investigated. Generally, this paper is written in high quality and easy to follow. Therefore, there are just a few minor suggestions to help further improve this work. Detailed comments are provided below.

1.        The meaning of symbol H in Eqs.(1) ~ (3) need to be clarified.

2.        How to consider the influence of sulphate-induced cracking on ionic transport, since the evolving porosity can only be less than the initial porosity as shown in Eq. (19).

3.        The determination procedure of the volume fraction of ITZ need to be clarified.

4.        Some recent developments on the competitive antagonism induced by the dual attack of chloride and sulfate are recommended to be included in the Introduction such as: https://doi.org/10.1016/j.conbuildmat.2021.122806

5.        The dash lines on the background of Fig. 3 are suggested to be removed to make the graph more concise. Same comment also applies to Figs. 3~7.

Author Response

The authors have revised our manuscript as per the reviewer's comments. Please find the attached revised manuscript and authors' response file.

Reviewer 3 Report

This research on modeling the ingress of sulphates into concrete offers some fresh perspectives and should be welcomed following extensive revisions to correct inaccuracies and acknowledge some of the severe limitations of the proposed approach.

The impression given by the paper that the authors are the first to use a multiphase model of concrete is not correct. For mechanical behaviors such models have been popular for several decades. The modeling of transport processes in concrete with multiphase models also has a long history. It is true that fewer such models have been used to simulate sulphate ingress but such models do exist as noted later in this set of comments.

The extensive set of parametric analyses given in the second half of the paper are presumptuous given the limitations of the model (as described later) and lack of rigor of the validations of the model. "resulting strain" plots are given but without that aspect of the model being validated. Service life results are also given but without that aspect of the model being validated.

Line 72. The paper claims " However, it should be emphasized here that a common limitation exists in these above models. Specifically, the modelled concrete is broadly hypothesized as a homogeneous matrix, rather than a multi-phase system comprising the bulk paste, aggregates and interfacial transition zones." This claim is not correct since the work of Idiart et al [27] simulates the concrete mechanical and sulphate transport processes using a multiphase model. The transport model accounts for two phases (the aggregates and matrix) since their interface is of zero thickness but nonetheless this is a multiphase model.

There are recent examples where all three phases (aggregates, matrix and interface) are present within the model. See for examples

++ Chen et al "A chemical-transport-mechanics.." Materials 2021; 14(24) 7710

++ Zhuang et al "Diffusion model of sulfate ions.. in mesoscopic numerical simulation" Structural Concrete 2022; doi.org/10.1002/suco.202100760

The merits of the authors model relative to these and the many other related models can be debated, but it is not correct to imply the authors are the first to implement a multi phase model especially given the multitude of previous works that employed three phase models of other forms of transport in concrete.

The choice of Voigt and Reuss theories to model the diffusivity of the three phase composite should be supported through more discussion noting some of the limitations. The paper says,  "Through this, the proposed model is able to capture the influence of each individual phase on the diffusion-reaction progress of sulphate ions in concrete" but the employed theories do not account of the spatial arrangement of the phases and how that locally affects transport or mechanical responses to sulphate attack. Near the concrete surface the arrangement of aggregates can be different than in the interior and such surface effects can be particularly important.

Figure 1 implies that the spatial arrangement of the phases is captured by the model when it is not. Also "paste" is not the correct term for what binds the coarse aggregates. Finer aggregates are also present.

Line 95. "the lifetime of concrete structures" is affected by numerous factors. It is inappropriate to imply a direct modeling of service life in this paper especially when considering that damage processes are not considered.

One of the main conclusions is that the aggregates are less diffusive and therefore act to impede sulphate ingress. Many previous models of chloride ingress, or ingress in general, have reached the same conclusion. This should not be highlighted as a new finding.

The approach represented by Equations 21 to 23 has some merit but strain based criterion does not account for the mechanics of fracture. Fracture is a process. Looking at line 392 "crack growth penetrates through the entire cover thickness of concrete" would be due to a process governed by fracture mechanics instead of a strain value at a point in space and a point in time.

Furthermore the approach by Equation 23 does not differentiate between the strength or strain capacities of the material phases. It is conceivable that expansive pressure could crack the weak interfaces and thus modify the transport process.

The work on validating the proposed approach looked only at concentration profiles of sulphate ingress. The paper did not attempt to validate the calculation of service life by Equation 23 or its strain component.

Line 431. As the authors noted earlier in the paper and several other studies have found porosity of the matrix can serve to accommodate a degree of the expansive processes such as those caused by sulphate exposure. The mention of "compactness" could therefore be misleading. The statement does not allow for an air void system with low diffusivity but with void space to accommodate expansion if it occurs.

The background discussion and the paper conclusions tend to imply that three phase modeling of concrete is an innovative feature of the authors' research. Three phase models including those for sulphate or chloride diffusion abound in the literature.  Any innovative aspects of the research should be more plainly described in the context of past work in this area.

The conclusions attempt to relate the rate of sulphate ingress to service life of concrete. Without including the processes of damage induced by expansive reactions or damage induced by other mechanisms the connections to service life more explanation. It is known that weak interfaces are susceptible to damage after which their role in transporting sulphates may increase. Many of the past modeling effects have included consideration of damage development and its affects on transport. This paper does not make this connection and therefore the stated conclusions on service life are tenuous.

Author Response

(The authors gave the same response as above.)

Round 2

Reviewer 3 Report

Several of the review comments have been adequately addressed through revision. The following comments still need attention.

As noted in the previous set of review comments, the model is based on some large simplifying assumptions. Notably, the processes of damage development are not modeled. Instead forecasts of cracking are based on diffusion calculations together with a model of volumetric expansion. As such, the statement in the abstract that the ITZ "plays a minor role in sulphate attack" is unwarranted, since microcracking along the interface could markedly change the modeled behavior.

The heading for section 3.1 should be modified to indicate the "validation" is for the diffusion profiles. Otherwise, one could be led to believe the whole of the modeling scheme presented in section 2, which included the durability forecasting, is being validated.

In the response statement, the authors rightly admit there is a scarcity of data on expansive strain due to sulphate attack and its relationships with service life. However, that does not mean the need for validation can be waived prior to running extensive parametric simulations. The authors should include brief discussion (at the start of section 3.2 and in the conclusions section) on the future needs for such data for validation purposes. It should be clear that the parametric simulation results supporting Figures 9, 10, and 11 are based on a model that is not sufficiently validated. It is important to note these validation needs exist.

Author Response

Dear editor and reviewer, please find the attached authors' response file.

Round 3

Reviewer 3 Report

The latest revisions are appreciated. They will improve the accuracy of the presentation of the research.